# The Current and Future Promises of Combination Radiation and Immunotherapy for Genitourinary Cancers

**DOI:** 10.3390/cancers15010127

**Published:** 2022-12-25

**Authors:** Ava Saidian, Isabella Dolendo, Andrew Sharabi, Tyler F. Stewart, Brent Rose, Rana R. McKay, Aditya Bagrodia, Amirali Salmasi

**Affiliations:** 1Department of Urology, University of California-San Diego, San Diego, CA 92121, USA; 2School of Medicine, University of California-San Diego, San Diego, CA 92093, USA; 3Department of Radiation Medicine and Applied Sciences, University of California-San Diego, San Diego, CA 92093, USA; 4Department of Medicine, University of California-San Diego, San Diego, CA 92161, USA; 5Moores Cancer Center, University of California-San Diego, San Diego, CA 92037, USA

**Keywords:** immunotherapy, radiotherapy, prostate cancer, bladder cancer, genitourinary malignancies

## Abstract

**Simple Summary:**

Contemporary immunotherapy agents have recently been approved for use in genitourinary malignancies. Combining immunotherapy with radiotherapy may have a synergistic effect in treating bladder and prostate cancer. This article reviews the available data on combination immunotherapy and radiotherapy in the treatment of prostate and bladder cancer.

**Abstract:**

As the indications for the use of immunotherapy in genitourinary malignancies expand, its role in combination with standard or conventional therapies has become the subject of contemporary studies. Radiotherapy has multiple immunomodulating effects on anti-tumor immune response, which highlights potential synergistic role with immunotherapy agents. We sought to review the body of published data studying the combination of immunotherapy and radiotherapy as well as the rationale for combination therapy. Trial information and primary articles were obtained using the following terms “immunotherapy”, “radiotherapy”, “prostate cancer”, and “bladder cancer.” All articles and trials were screened to ensure they included combination radiotherapy and immunotherapy. The effects of radiation on the immune system, including both immunogenic and immunosuppressive effects, have been reported. There is a potential for combinatorial or synergistic effects between radiation therapy and immunotherapy in treating bladder and prostate cancers. However, results from ongoing and future clinical trials are needed to best integrate immunotherapy into current standard of care treatments for GU cancers.

## 1. Introduction

Bladder and prostate cancer are among the most morbid and common malignancies in the US, respectively. Radiation therapy has been well-defined in the treatment of both malignancies. Radiotherapy is one of the principal strategies for bladder-sparing trimodal therapy for the treatment of bladder cancer. In the realm of prostate cancer, radiotherapy is utilized in treatment of localized and metastatic disease with curative, adjuvant, and palliative indications [1]. The role of immunotherapy in genitourinary malignancies has widely expanded in the past decade. Multiple checkpoint inhibitors have been approved for advanced bladder cancer including frontline use, maintenance post chemotherapy, and as second-line treatment. Additionally, immunotherapy combinations are being explored across all advanced disease states. Nivolumab was recently approved for the treatment of individuals with post-cystectomy bladder cancer who are at high risk for recurrence [2]. In contrast to bladder cancer, immunotherapy has a limited role in management of prostate cancer. While sipuleucel-T was the first autologous vaccine to prolong survival for prostate cancer, subsequent unselected immunotherapy strategies have been largely unsuccessful [3,4,5]. The only current indication for immune checkpoint inhibitors is in patients with high tumor mutational burden or microsatellite instability [6,7]. Along with approval of these agents, there is a growing body of evidence highlighting the potential synergistic role of combination immunotherapy and radiotherapy. In this article, we review the immunomodulating effects of radiation and its enhancement when combined with immunotherapy, and we report on the current body of data regarding combination radiation treatment and immunotherapy in prostate and bladder cancer.

## 2. Materials and Methods

The goal of this narrative review is to identify and summarize the previously published and on-going studies in combined radiation and immunotherapy in the treatment of bladder and prostate cancer. A search was conducted on PubMed and ClinicalTrials.gov to identify relevant literature and trials. The key words “bladder cancer”, “prostate cancer”, “radiotherapy”, and “immunotherapy” were used to conduct our initial inquiry. A total of 23 unique bladder cancer trials and 14 prostate cancer trials that included combination radiotherapy and immunotherapy were found and included in our review.

## 3. Effects of Radiotherapy on the Immune System

Radiotherapy has multiple immunomodulating effects. It can induce immunogenic cell death, enhance immunogenic antigen presentation, and activate anti-tumor immune response [8]. It mediates cell death primarily via photons that damage DNA, inducing double-stranded breaks, and by generating hydroxyl free radicals [9]. Radiotherapy has also been shown to cause immunogenic cell death by activating the immune system against tumor cells and inducing a proinflammatory microenvironment [9,10]. This effect is achieved by multiple mechanisms. Firstly, tumor antigens that are typically concealed from the immune system are released during cell death, allowing antigen presenting cells to cross present tumor specific antigens to T-cells [11,12]. By increasing the exposure of tumor-specific antigens to the immune system, radiation can help stimulate an effective anti-tumor T cell response, termed an in situ vaccination effect [13,14,15].

In addition, the release of damage-associated molecular patterns (DAMPs), including calreticulin, a phagocytosis-promoting protein, further promotes the immune response [8,16]. When dendritic cells and antigen-presenting cells are activated, they undergo maturation and then travel to lymph nodes to prime T cells, which infiltrate into tumors [8,12,17].

Radiation has also been shown to upregulate the cell surface expression of MHC class I in a dose-dependent manner mediated by radiation-induced upregulation of mTOR, further facilitating CD8+ T cell priming [18]. Treating tumors with radiation prior to immunotherapies could enhance tumor susceptibility to immunotherapies by priming a systemic anti-tumor immune response [19,20]. Adhesion molecules such as ICAM1 and VCAM-1 on tumor vessels are also upregulated, leading to increased extravasation of T cells to tumor sites [21,22]. Radiotherapy further modifies the tumor microenvironment by upregulating certain chemokines that promote recruitment of T cells, including CXCL9, CXCL10, and CXCL16 [23,24,25,26]. Other proinflammatory cytokines, such as TNFα, IL-1β, and type 1 and 2 interferons, are also increased [25,27,28,29,30]. The cGAS-STING pathway mediates the activation of type 1 interferon, inducing recruitment of effector T cells and antigen presenting cells [31]. Radiation treatment also increases NKG2D ligands, activating NK cell-mediated cytolysis [32,33]. This multifaceted interaction of radiation with the immune system creates an anti-tumorigenic microenvironment that could act synergistically with immunotherapy.

In addition to its local effects, radiotherapy has also been described to have an abscopal effect, the rare phenomenon of anti-tumor responses in areas outside of the primary local irradiation site [34,35,36,37]. Although quite infrequent, triggering systemic abscopal effects through combination immunotherapy and radiation has been described by multiple groups [34]. It is hypothesized to be mediated by a systemic anti-tumor immunological response; therefore, it is postulated that the addition of immunotherapy to radiation could enhance the abscopal response, affecting distant areas and reducing overall tumor burden [38,39]. Abscopal effects at the microscopic level could have the added benefit of preventing disease recurrence by targeting microscopic seeds of metastasis [10]. 

While there are potential beneficial effects of radiation therapy, the effects of radiation on the immune system are complex and can also have immunosuppressive effects. For example, radiation can have bystander effects and directly inducing DNA breaks and apoptosis in local immune cells as well as mediate other immunosuppressive effects, decreasing the immunogenic responses to radiotherapy [40].

Radiation-induced lymphopenia occurs in 40–70% of patients undergoing conventional external-beam radiation [41]. Specifically in prostate cancer patients, higher rates of radiation-induced lymphopenia have been observed with pelvic nodal irradiation compared to those without [42]. Studies have shown that radiation-induced lymphopenia serves as a negative prognostic factor in some cancers [43,44]. In a study of muscle-invasive bladder cancer patients, patients with recurrent or residual tumors had significantly lower rates of recovery from radiation-induced lymphopenia compared to patients who were disease-free for five years [45]. Radiotherapy has also been shown to stimulate myeloid-derived suppressor cells (MDSCs), which are key mediators of immunosuppression by inhibiting effector T cells and inducing regulatory T cells, thus promoting tumor progression and local invasion [46,47,48,49].

One strategy to enhance the activity of radiation is to block the immunosuppressive effects of radiation and harness the pro-immunogenic effects. We further explore specific immunosuppressive effects of radiotherapy and how checkpoint blockade immunotherapies can be used to target and reverse such effects. PD-L1 has increased expression in response to radiation treatment, particularly in urothelial muscle-invasive bladder cancer [50,51]. This inhibits the cell mediated immune response by binding PD-1 receptors of tumor-specific T cells in lymph nodes [50,52]. The addition of PD-1 blockade after radiation has been shown to overcome adaptive resistance to systemic anti-tumor immunity via the PD-L1 pathway, improving local and distal tumor control [53]. In a mouse model of urothelial muscle-invasive bladder cancer, the addition of PD-1 blockade to radiation resulted in a significantly slower growth rate and improved survival compared to radiation alone [54]. Slower growth rates were observed not only in the irradiated tumors but also in the contralateral non-irradiated tumors [54]. Similarly, in a mouse model of castration-resistant prostate cancer, the combination of radiation and checkpoint inhibitors led significant increase in survival in radiation treatment combined with either anti-PD1 or anti-PDL1 compared to monotherapy. Additional treatment with anti-CD8 antibody blocked the survival effect. An abscopal treatment effect was observed in radiation group [55].

Another study found that patients with oligometastatic solid tumors were treated with Sunitinib, and radiation reduced accumulation and the immune-suppressive function of MDSCs and exhibited improved progression-free survival in comparison to those only treated with radiation [47]. Radiotherapy has also been shown to upregulate cytotoxic T lymphocyte antigen 4 (CTLA4) expressed on regulatory T cells and compete with effector T cells to bind dendritic cells, thus dampening the immune response [56,57]. A study looked at the combined use of radiation and the CTLA4 inhibitor ipilimumab in patients with advanced melanoma. Koller et al. found that compared to ipilimumab alone, the combined use of ipilimumab and radiation resulted in better median overall survival and complete response rate, but no improvement in progression-free survival was observed [58]. 

Additionally, regulatory T cells, which have been thought to play a role in anti-tumor immunity, have shown to be more resistant to ionizing radiation compared to other immune cell types, thus leading to their preferential proliferation [12,59,60,61]. It has been shown that depleting immunosuppressive Foxp3+ regulatory T cells enables increased T cell activation, infiltration, and tumor destruction [62]. Blocking the migration of regulatory T cells could also be an effective way to block the immunosuppressive effects of proliferating regulatory T cells [63]. The cGAS-STING pathway that induced type 1 interferon can also provide an immunosuppressive effect by upregulating transforming growth factor β (TGFβ), an anti-inflammatory cytokine [64,65,66]. TGFβ and specific stromal fibroblast populations contribute to decreased response to anti-PD-L1 agents in patients with urothelial cancer; therefore, TGFβ could provide a key target for inhibition to enhance both immunotherapy and radiotherapy responses [67,68].

The clinical benefits of combination radiation with immunotherapy have been shown in metastatic solid tumors, non-small-cell lung cancer, and metastatic renal cell carcinoma [69,70]. Despite the reported benefits of combination radiation with immunotherapy for cancer treatment, there are many questions about the optimal application of the combination remains unanswered: the questions about the ideal radiation field size, radiation dose, and fractionation, as well as ideal sequencing and immunotherapy agent [69]. While some preclinical and clinical studies have shown a more robust immune response after fractionation radiation compared with single-dose radiation, other studies have demonstrated no benefit from fractionation versus single-dose radiation [21,39,71,72,73,74]. The current data also support simultaneous administration of radiation and immunotherapy for maximal anti-cancer activity. However, further studies needed to establish the optimal timing of radiation protocol based on mechanism of immunotherapy agents [8,39,75,76].

## 4. Radiotherapy in Combination with Immunotherapy in Bladder Cancer

Traditionally, the gold standard for treatment of muscle invasive bladder cancer (MIBC) was limited to chemotherapy and radical cystectomy. Recently, bladder-sparing trimodal therapy (transurethral bladder tumor resection followed by radiation and concurrent chemotherapy) has become a treatment option for MIBC [77]. Combination immunotherapy with radiation therapy in the management of bladder cancer is the primary subject of several ongoing clinical trials (Table 1). 

A phase II trial (NCT02662062) demonstrated satisfactory safety and promising efficacy of chemoradiotherapy (64 Gy in 32 daily radiation fractions) in combination with 6 weekly doses of cisplatin and concurrent pembrolizumab (200 mg IV q21 days for 7 doses) in 10 patients with MIBC [77]. The primary endpoint was feasibility, defined by a satisfactory low rate of unacceptable toxicity of grade 3 or 4 non-urinary adverse events or failure of completion of planned radiation therapy according to defined parameters. One patient had a dose of cisplatin withheld. Four of the ten patients experienced G3 – 4 non-urinary adverse events within 12 weeks of completing treatment. One immune-related adverse event interrupted pembrolizumab delivery (G2 nephritis). By week 24, 9/10 patients achieved a complete cystoscopic response to treatment and were free of distant metastatic disease. A similar multicenter phase II trial (NCT02621151) evaluated the safety and efficacy of pembrolizumab in addition to trimodal bladder preservation therapy (TMT) [78]. This study population was divided into a safety cohort (SC) and efficacy cohort (EC). Patients received pembrolizumab 200 mg × 1 followed 2–3 weeks by maximal TURBT and then whole bladder radiation (52 Gy/20 fx; IMRT preferred) with twice weekly gemcitabine 27 mg/m^2^ and pembrolizumab every 3 weeks for three treatments. The primary endpoint was 2 y bladder-intact disease-free survival (BIDFS: first of MIBC or regional nodal recurrence, distant metastases, or death) assessed by serial cystoscopy/cytology and CT/MRI. The estimated 1-year BIDFS rate is 77% (95% CI: 0.60–0.87). Twelve-week complete response rate was 100% in SC and 83% for EC. In the EC, 35% of patients had a ≥3 treatment-related adverse events (grade 3 events included UTI 8%, diarrhea 4%, colitis 4%, bladder pain/obstruction 4%, neutropenia 2%, and thrombocytopenia 2%). Pembrolizumab-related grade ≥3 adverse events included three patients (6%) with GI toxicity, of which one patient had a colonic perforation. One patient died due to fungemia, unrelated to the study therapy.

Ongoing studies are examining the role of combining immunotherapy with radiation for patients with bladder cancer. The phase II NUTRA trial (NCT03421652) is currently enrolling patients with non-MIBC for chemotherapy or cystectomy and administering nivolumab (240 mg IV q2 weeks for a maximum of 6 months) concurrently with SOC radiation therapy for bladder cancer [79]. A total radiation dose of 64 gray in 32 fractions was administered per standard of care for bladder cancer. If local lymph nodes were clinically involved, they had to be radiated. The primary outcome of the study is progression-free survival at 12 months and has yet to be reported. However, 6 of 14 patients have demonstrated a completed response: 4 had residual disease and 4 had disease progression. Nivolumab and radiation therapy toxicities were as expected: five patients needed steroids due to immune-mediated adverse events; diarrhea was observed in two patients; thyroid dysfunction was observed in two patients; and immune cystitis in was observed in one patient. No treatment related deaths were noted. A phase II trial of durvalumab plus tremelimumab with concurrent radiotherapy preliminary reported safety and efficacy of combination treatment in patients with MIBC [80]. Treatment consisted of initial TURBT followed by durvalumab 1500 mg i.v. plus tremelimumab 75 mg i.v., every 4 weeks for three doses. Normofractionated external-beam radiation was started 2 weeks later, at doses of 46 Gy to minor pelvis and 64–66 Gy to bladder. A complete response at post-treatment biopsy was documented in 26 (81%) patients: 2 patients had residual MIBC, and 4 patients were not evaluated due to rejection, clinical impairment, death from COVID 19, and a suspected treatment-related death from peritonitis (one each). After a median follow up of 6.1 months (2.5–20.1), two patients underwent salvage cystectomy because of MIBC and T1 relapses. The estimated 6-month rates for disease-free survival (DFS) with bladder intact, DFS, and overall survival were 76% (95%CI, 61–5%), 80% (95%CI, 66–98%) and 93% (95%CI, 85–100%),. A total of 31 (97%) patients experienced adverse events related to radiation and/or immunotherapy, with diarrhea (41%) and urinary disorders (37.5%) as the most frequent. Grade 3 or 4 adverse events related to therapy were reported in 31% of patients, the most frequent being gastrointestinal toxicity (12.5%), acute kidney failure (6%), and hepatitis (6%). In another ongoing trial (NCT04936230), Himanshu et al. are evaluating the effect of stereotactic body radiation therapy (3 fractions over 2 weeks) in combination with atezolizumab in platinum ineligible/refractory metastatic urothelial cancer. 

These results highlight the potential for radiation to synergize with immunotherapy in treatment of MIBC. However, checkpoint blockade immunotherapies are not intrinsic radiosensitizers and do not function primarily to enhance DNA damage from radiation. Currently, platinum-based chemotherapies remain the most effective radiosensitizers known, and, thus, caution should be taken when trying to replace platinum-based chemotherapies with immunotherapies. Alternatively, the use of immunotherapy after completion of a course of radiation or chemoradiation may be an advantageous strategy to enhance immune-mediated clearance after maximal tumor cytoreduction. The results from ongoing clinical trials are highly anticipated and will provide clinical evidence on how to best integrate immunotherapies with the current conventional modalities. 

## 5. Radiotherapy in Combination with Immunotherapy in Prostate Cancer

There are a myriad of ongoing trials studying the role of combined radiotherapy and immunotherapy in prostate cancer (Table 2). However, only a few RCTs have published results. A phase II trial randomized 49 patients with mCRPC to either sipuleucel-T alone or sipuleucel-T preceded by external-beam radiation therapy [63]. There was no statistically significant difference in progression free survival (*p* = 0.06) between the combination EBRT + sipuleucel-T arm (3.65 months) and the sipuleucel-T only arm (2.46 months). A single-arm phase II trial evaluated the utility of stereotactic ablative radiotherapy in addition to sipuleucel-T in patients with mCRPC. The median time to progression was 1.2 weeks (95% CI 6.8–14.0 weeks). [64] Unlike the phase II randomized trial combining EBRT and sipuleucel-T, this regimen did induce humoral and cellular immune responses. However, the immune response induced by stereotactic ablative radiotherapy did not yield a clinical benefit compared to previously reported outcomes of patients treated with sipuleucel-T.

A multicenter, randomized, double-blind phase III trial for men with mCRPC (with at least one bone metastasis) who had progressed on docetaxel randomized radiotherapy followed by ipilimumab (10 mg/kg q 3 weeks) vs. radiotherapy followed by placebo was reported by Kwon ED. et al [62]. The study was powered for a primary endpoint of overall survival. Unfortunately, in the initially report there was no significant difference in median overall survival of 11.2 months (95% CI 9.5–12.7) in the experimental arm versus 10.0 months (95% CI 8.3–11.0) in the placebo group; however, the *p* value was very close to significance at *p* = 0.053. In a subsequent subset analysis, there appeared to be a meaningful benefit to ipilimumab in men with a good performance status and absence of visceral metastases. Furthermore, in a recent pre-planned final analysis of this phase III trial, there was an excess of long-term survivors in the radiation plus ipilimumab arm with overall survival rates at 3 years and 5 years approximately two to three times higher in the radiotherapy + ipilimumab arm [81]. These data indicate that checkpoint blockade immunotherapies may have activity in a subset of men with prostate cancer and that additional biomarkers may be needed to guide precision medicine and identify those patients who are most likely to benefit from combinatorial therapies. 

## 6. Conclusions

Radiation therapy remains one of the most effective anti-cancer treatments available and is part of the standard of care for many different types of cancer. The effects of radiation on the immune system, including both immunogenic and immunosuppressive effects, have been reported. Thus, there is a potential for combinatorial or synergistic effects between RT and immunotherapy in treating GU malignancies. However, results from ongoing and future clinical trials are needed to help guide ideal radiation field size, RT dose and fractionation, as well as ideal sequencing to best integrate immunotherapy into the current standard of care treatments for GU cancers.

## Figures and Tables

**Table 1 cancers-15-00127-t001:** Clinical Trials of Immunotherapy and Radiotherapy in Bladder Cancer.

Study	Phase	Intervention	Patient Stage	Status
NCT03950362 [PREVERT]	II	Avelumab + RT	<pT2 N0M0	Not yet recruiting
NCT03529890 [RACE-IT]	II	Nivolumab + RT + radical cystectomy with pelvic lymphadenectomy	cT3 –T4 cN0/N + cM0	Active, not recruiting
NCT05445648 [CBPTMI]	II	Tislelizumab + TURBT + RT	cT2 –T4a N0M0	Not yet recruiting
NCT04543110 [RADIANT]	II	Durvalumab + RT	cT2 –T4a N0M0	Recruiting
NCT03702179 [IMMUNOPRESERVE]	II	NCT04216290 + tremelimumab + RT	cT2 –T4a N0M0	Active, not recruiting
NCT03747419	II	Avelumab + RT	≥pT2, cN0M0	Recruiting
NCT04216290 [INSPIRE]	II	durvalumab + RT + chemotherapy	Any T, any N, M0	Recruiting
NCT02560636 [PLUMMB]	I	Pembrolizumab + RT	T2 –4, N0 –3, M0 –1	Recruiting
NCT04902040	I/II	Plinabulin + RT+ atezolizumab or Avelumab or durvalumab or Nivolumab or Pembrolizumab	Any T, any N, M+	Recruiting
NCT04936230	II	Atezolizumab + SBRT	Any T, any N, pM+	Recruiting
NCT03617913	II	Avelumab + RT + cisplatin chemotherapy	pT2 –T4a N0M0	Completed
NCT03697850 [BladderSpar]	II	Atezolizumab + chemo-radiotherapy	pT2 –T3 cM0	Recruiting
NCT02621151	II	Pembrolizumab + EBRT + gemcitabine + TURBT	T2 –T4a, N0M0	Active, not recruiting
NCT03693014	II	SBRT + ipilimumab + nivolumab + pembrolizumab + atezolizumab	Any T, any N, M+	Recruiting
NCT03775265	III	Atezolizumab + chemoradiotherapy	T2 –T4a N0M0	Recruiting
NCT05531123	II	Tislelizumab + modified hypofractionation + gemcitabine and cisplatin	cT2 –4a, N0 –1, M0	Not yet recruiting
NCT05241340 [RAD-VACCINE]	II	Sasanlimab + SBRT + radical cystectomy	cT2 –4a N0M0	Recruiting
NCT05259319 [IMMUNOs-SBRT]	I	Atezolizumab + tiragolumab + SBRT	cM+	Not yet recruiting
NCT03915678 [AGADIR]	II	Atezolizumab + BDB001 (toll-like receptor agonist) + RT	cM+	Recruiting
NCT03589339	I	Nivolumab or pembrolizumab + SMRT	cM+ (<5 lesions)	Recruiting
NCT04977453	I/II	GI-101 + RT	“Advanced and/or metastatic”	Recruiting
NCT04241185 [KEYNOTE-992]	III	Pembrolizumab + RT + ciplatin + 5-FU + Mytomycin C + gemcitabine vs. Placebo to pembrolizumab	cT2 –T4, N0M0	Recruiting
NCT03768570	II	Trimodality therapy +/- durvalumab	cT2 –T4 N0M0	Recruiting

**Table 2 cancers-15-00127-t002:** Clinical Trials of Immunotherapy and Radiotherapy in Prostate Cancer.

Study	Phase	Intervention	Patient Population	Status
NCT01436968 [PrTK03]	III	Aglatimagene besadenovec + valacyclovir + standard RT	Intermediate or high risk (1 high risk feature), M0	Active, not recruiting
NCT02107430	II	Dendritic cells DCVAC/PCa + standard RT	High or very high risk	Completed
NCT03543189	I/II	Nivolumab + brachytherapy + EBRT	Grade group 5, any PSA or T stage	Recruiting
NCT01807065	II	Sipuleucel-T + EBRT	mCRPC	Completed
NCT03795207 [POSTCARD]	II	Durvalumab + SBRT	Biochemical recurrence (BCR), M0	Recruiting
NCT05361798	II	Immunocytokine M9241 + SBRT	BCR, ≤5 bone or LN metastases	Recruiting
NCT01818986	II	Sipuleucel-T + SBRT	mCRPC	Completed
NCT04071236	I/II	Avelumab + radium Ra 223 dichloride	mCRPC	Recruiting
NCT02232230	Retrospective observational	Provenge + RT	mCRPC	Completed
NCT03007732	II	Pembrolizumab + SBRT +/- intratumoral SD-101	mCSPC	Recruiting
NCT00005916	II	PSA-Based Vaccine + RT	Treatment naïve local disease	Completed
NCT04946370	I/II	225Ac-J591 (a drug that can deliver radiation to prostate cancer cells) + pembrolizumab	mCRPC	Recruiting
NCT03217747	I/II	Avelumab + utomilumab + RT	mCRPC	Active, not recruiting
NCT02463799	II	Radium-223 + sipuleucel-T	mCRPC	Completed

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
