# Peer review of "The Current and Future Promises of Combination Radiation and Immunotherapy for Genitourinary Cancers"

_cancers, 2022, doi:10.3390/cancers15010127_

Round 1

Reviewer 1 Report

This article summarized combination of radiotherapy and immunotherapy in bladder and prostate cancers. The authors well summarized clinical trials of radiotherapy and immunotherapy in bladder and prostate cancers. However, this article has several significant flaws.

Major

1) I’m not sure if this paper is a systematic review, narrative review, or expert opinion.

2) The author described a few methods to review papers only in the abstract. The materials and methods should be described in details.

3) This paper seems to review only clinical trials. Many retrospective papers have already reported the outcomes of combination of radiotherapy and immunotherapy. The authors should summarize these papers.

Author Response

Thank you for taking time to review our manuscript. We appreciate your comments and have taken them in to consideration. Please see our responses below. 

  1. Narrative review- this has now been clarified in our paper.
  2. A materials and methods section has been added.
  3. Retrospective papers are cited in our manuscript. The trials they included have been summarized in our paper.

Reviewer 2 Report

1.     Very similar review was already published in Cancers last year as shown below.  If possible, in order to differentiate the present study from the previous review, the authors should include the studies for metastatic UC patients.

Bladder-Sparing Chemoradiotherapy Combined with Immune Checkpoint Inhibition for Locally Advanced Urothelial Bladder Cancer-A Review. van Hattum JW, de Ruiter BM, Oddens JR, Hulshof MCCM, de Reijke TM, Bins AD.  Cancers (Basel). 2021 Dec 22;14(1):38. doi: 10.3390/cancers14010038.

2.     Page 2, line3 

The only indication for c.  I wonder what c is.

Author Response

Thank you for taking time to review our manuscript. We appreciate your comments and have taken them in to consideration. Please see our responses below. 

  1. Our review includes trials enrolling patients with metastatic UC.

2. Corrected to "immune checkpoint inhibitors"

Reviewer 3 Report

This work is an overview of current evidence regarding combination of immunotherapy and radiotherapy in bladder and prostate cancer. The topic is interesting and the study could be considered for publication should the authors address the following: 

1. The manuscript needs linguistic improvements by a medical writer or a language-edditing software. It cannot be accepted in the current form. 

2. The authors should inform the readers about the staging groups of the patients in the included studies (eg. locally advanced - non-metastatic, oligometastatic etc). Where all patients in the primary setting or also in recurrence? 

Author Response

Thank you for taking time to review our manuscript. We appreciate your comments and have taken them in to consideration. Please see our responses below. 

  1. We revised our manuscript using editing software with a final score of 100% in English language clarity, conciseness, punctation conventions and vocabulary. It was also reviewed by a native english speaker

2. The staging groups of the patients has been added to the manuscript. 

Round 2

Reviewer 1 Report

The authors answered that retrospective papers are cited and the trials they included have been summarized in the paper. I checked the manuscript, but I highly doubt that an adequate literature search and review has been done. This paper shows only ongoing clinical trials. The authors should do a literature search and summarize retrospective studies as well as clinical trials.

Author Response

We reviewed the literature and there were no retrospective studies evaluating immunotherapy + RT in bladder or prostate cancer in humans (mouse model studies were published). There are review articles on the topic. The only studies in these reviews were analysis of the clinical trials that we have also reviewed. We would appreciate further direction if we are missing any major retrospective studies. 

Reviewer 3 Report

I have no further comments. The study is suitable for publication 

Author Response

Thank you

Round 3

Reviewer 1 Report

I recommend acceptance of this paper.